# Rapid Detection of *Aspergillus flavus* and Quantitative Determination of Aflatoxin B_1_ in Grain Crops Using a Portable Raman Spectrometer Combined with Colloidal Au Nanoparticles

**DOI:** 10.3390/molecules27165280

**Published:** 2022-08-18

**Authors:** Huiqin Wang, Mengjia Liu, Yumiao Zhang, Huimin Zhao, Wenjing Lu, Taifeng Lin, Ping Zhang, Dawei Zheng

**Affiliations:** Faculty of Environment and Life, Beijing University of Technology, Beijing 100124, China

**Keywords:** *Aspergillus flavus*, aflatoxins B_1_, carcinogen, Surface-Enhanced Raman Scattering, colloidal Au nanoparticles, portable Raman spectrometer

## Abstract

*Aspergillus flavus* and Aflatoxins in grain crops give rise to a serious threat to food security and cause huge economic losses. In particular, aflatoxin B_1_ has been identified as a Class I carcinogen to humans by the International Agency for Research on Cancer (IARC). Compared with conventional methods, Surface-Enhanced Raman Scattering (SERS) has paved the way for the detection of *Aspergillus flavus* and Aflatoxins in grain crops as it is a rapid, nondestructive, and sensitive analytical method. In this work, the rapid detection of *Aspergillus flavus* and quantification of Aflatoxin B_1_ in grain crops were performed by using a portable Raman spectrometer combined with colloidal Au nanoparticles (AuNPs). With the increase of the concentration of *Aspergillus flavus* spore suspension in the range of 10^2^–10^8^ CFU/mL, the better the combination of *Aspergillus flavus* spores and AuNPs, the better the enhancement effect of AuNPs solution on the *Aspergillus flavus*. A series of different concentrations of aflatoxin B_1_ methanol solution combined with AuNPs were determined based on SERS and their spectra were similar to that of solid powder. Moreover, the characteristic peak increased gradually with the increase of concentration in the range of 0.0005–0.01 mg/L and the determination limit was 0.0005 mg/L, which was verified by HPLC in ppM concentration. This rapid detection method can greatly shorten the detection time from several hours or even tens of hours to a few minutes, which can help to take effective measures to avoid causing large economic losses.

## 1. Introduction

*Aspergillus flavus*, a common saprophytic mold widely existing all over the world, has been recognized as the main pathogenic fungus causing grain crops mildew in the process of storage. *Aspergillus flavus* and the closely related subspecies *Aspergillus parasiticus* can contaminate grain crops in a wide range [1]. According to FAO reports, the annual losses caused by fungal pollution in the world have reached tens of billions of dollars, and most of them are caused by *Aspergillus flavus* contamination [2]. They can consume a lot of nutrients, accelerate fat deterioration, and destroy protein, pantothenic acid, niacin, vitamin A, vitamin D, vitamin E, and other components, resulting in the nutritional reduction of grain crops. Moreover, 30–60% of them can produce aflatoxins under appropriate conditions [3].

Aflatoxins are cancerous secondary metabolites from *Aspergillus flavus* and *Aspergillus parasiticus* [4]. They are toxic to humans and animals causing liver damage, abnormalities, mutations, and cancer, and when in high doses, aflatoxins can be fatal [5]. Due to their high toxicity and carcinogenic potential, they are a high concern for the safety of food worldwide [6]. At present, more than 20 species of aflatoxins have been found, mainly including B_1_, B_2_, G_1_, G_2_, M_1_, M_2_, etc. Among them, aflatoxin B_1_ is the most toxic and carcinogenic and has been identified as a Class I carcinogen to humans by the International Agency for Research on Cancer (IARC) [7,8]. Given these adverse effects, regulatory control limitations for aflatoxin B_1_ in food and feed are well-established. China has strictly set limits of 20 μg/kg of aflatoxin B_1_ in corn and corn products, according to the national food safety standard (GB 2761-2017) [9].

Several methods and techniques for the detection and quantification of *Aspergillus flavus* and Aflatoxins have been developed. DNA-based techniques have been used for the detection of aflatoxigenic strains of *Aspergillus flavus*. They mainly include the plate count method [10], polymerase chain reaction (PCR) and Quantitative PCR (qPCR) [11,12,13,14], enzyme-linked immunosorbent assay (ELISA) [15,16,17,18], and so on. High-performance liquid chromatography (HPLC) [19], thin layer chromatography (TLC) [20], enzyme-linked immunosorbent assay (ELISA) [21], and electrochemical impedance spectroscopy (EIS) [22,23] methods have been used for the detection of aflatoxins. In recent years, molecular spectrum and hyperspectral imaging were widely used to detect *Aspergillus flavus* and Aflatoxins [24]. Raman spectroscopy (RS), a modern analytical technique that provides information about molecular vibrations and consequently the structure of the analyzed specimen, has been broadly used in various research fields ranging from the medical biological field [25,26,27,28] to food safety [29,30] and electrochemistry [31].

Surface-enhanced Raman scattering (SERS) based on noble metal nanomaterials or rough surface of a metal sheet has attracted increasing attention due to its unique characteristics of high sensitivity and the capability of chemical fingerprint recognition. Caldwell et al. utilized spherical gold nanoparticles with 14 nm and 46 nm diameters to improve the scattering signal obtained during Raman spectroscopy measurements to detect small plastic particles [32]. Bharathi et al. utilized picosecond laser-ablated gold nanoparticles (Au NPs) as surface-enhanced Raman scattering (SERS) substrates to detect the dye methylene blue and a chemical warfare agent simulant (methyl salicylate) [33]. Zavyalova et al. provided a SERSaptasensor based on colloidal solutions, which combines rapidity and specificity in the quantitative determination of the SARS-CoV-2 virus [34]. Compared with the colloidal solutions, there is greater SERS signal stability and a better detection limit may be achieved that allows the detection of low concentrations up to single-molecule level based on some SERS substrates that were prepared by the template-assisted electrodeposition [35], binary-template-assisted electrodeposition [36], pulsed laser ablation [37], and other methods. The surface nanostructures of artificially roughened metal thin films display many hot spots making them excellent SERS substrates [38]. The conventional approaches have several limitations including complicated pretreatment steps, requiring expensive instruments, operational complexity, lack of instrument portability, and difficulties in real-time monitoring [39,40,41]. Due to the toxicity of aflatoxin and people’s attention to food safety, more and more studies on using SERS technology to detect food security have appeared. The determination of aflatoxin B_1_ in peanut based on QuEChERS purification and surface-enhanced Raman spectroscopy (SERS) was carried out by Wang et al. [42]. Yang et al. used Raman spectroscopy technology to detect zearalenone (ZEN) and aflatoxin B_1_ in six kinds of maize samples with different mold degrees [43]. Therefore, it is of great significance to develop a fast, solvent-free, and cost-effective analytical method for noninvasive, rapid, and sensitive detection of *Aspergillus flavus* and Aflatoxin B_1_ in grain crops to prevent potential economic losses.

An important application of SERS in pathogenic microorganisms is to rapidly detect and identify pathogenic bacteria directly isolated from samples without relying on a culture medium, so as to improve efficiency and reduce cost. In this paper, the rapid detection of *Aspergillus flavus* and quantification of Aflatoxin B_1_ in grain crops using a portable Raman spectrometer-based colloidal Au nanoparticles (AuNPs) will be presented. Detection results of Aflatoxin B_1_ in grain crops were verified by HPLC in ppM concentration. The characteristics of *Aspergillus flavus* and Aflatoxin B_1_ will help to identify the degree of contamination by nondestructive testing of grain crops and gain timely control. In addition to the diagnostic application, this method is also potentially helpful for further determining the storage methods of grain crops.

## 2. Results and Discussion

### 2.1. SERS of Aspergillus flavus on the Medium and on Corn

A large number of biochemical components on the cell membrane surface of pathogenic microorganisms can be regarded as the characteristic signs of microorganisms. The structure and chemical composition information of the substance can be obtained based on SERS at the single-molecule level. Therefore, SERS has fingerprint recognition characteristics and high detection sensitivity of surface species, which can be used as a sign for the rapid identification and identification of fungi.

When *Aspergillus flavus* was inoculated on the culture medium, the increase of *Aspergillus flavus* on the culture medium could be clearly observed with the naked eyes with the increase of culture days. As shown in Figure 1, the surface color of the medium inoculated with *Aspergillus flavus* spores changed significantly from transparent color to black with the increase in time. The picture of day 0 showed the fresh medium, which was distinct from the following pictures of day 1, day 2, and day 3, respectively. After centrifuging the culture medium (1 mL) at 8000 *g* for 5 min, the *Aspergillus flavus* spores were collected and resuspended in 1 mL of 0.85% sterile normal saline, and then centrifuged and washed under the same conditions. The above procedure was repeated another 2–3 times to remove the culture medium and get the sample, of which 20 μL was taken and added to 500 μL of AuNPs solution and mixed for SERS detection. The SERS of *Aspergillus flavus* had fingerprints at 400–1800 cm^−1^, as shown in Figure 2a, which were mainly reflected in the characteristic peak absorption and spectral shape, especially at the range of 600–800 cm^−1^, 1200–1400 cm^−1^, and 1500–1600 cm^−1^. The peak of the surface-enhanced Raman spectra of *Aspergillus flavus* was analyzed and explained. The peak at 1605–1615 cm^−1^ represents C=O stretching in proteins [44], 1343–1346 cm^−1^ represents DNA base [45], 1315–1317 cm^−1^ represents the vibration of (–C=C–) conjugated of Amine III [26], 1302–1306 cm^−1^ represents carbohydrates [44], and 805–825 cm^−1^ represents protein, respectively [46]. The intensity of the Raman characteristic peak at 1535–1537 cm^−1^ increased obviously with the growth of culture time of *Aspergillus flavus* spore suspension. The experimental results shown from the SERS of *Aspergillus flavus* were consistent with the color changes observed directly on the culture medium. The comparison of non-SERS and SERS of *Aspergillus flavus* was shown in Figure 2b. The results showed that the SERS signals of *Aspergillus flavus* would increase with the coupling with AuNPs, while under non-SERS conditions, the Raman signals of normal *Aspergillus flavus* could hardly be detected by a portable Raman spectrometer.

When *Aspergillus flavus* was inoculated on corn grains, the amount of *Aspergillus flavus* gradually increased and the color of corn changed significantly with the increase in time, as shown in Figure 3. The SERS of *Aspergillus flavus* inoculated on corn grains shown in Figure 4 was similar to that inoculated on the medium, and they had the same fingerprints at 400–1800 cm^−1^. Moreover, the intensity of Raman characteristic peak at 1536–1537 cm^−1^ increased obviously with the growth of culture time of Aspergillus flavus on corn in different culture periods. The experimental results shown from the SERS of Aspergillus flavus were consistent with the color changes observed directly on corn.

### 2.2. SERS of Aspergillus flavus with Different Concentrations

Colloidal Au nanoparticles (AuNPs) with a particle size of about 60 nm, prepared according to a previously published procedure [47], served as SERS substrates in this work. *Aspergillus flavus* with different concentrations were added to a certain amount of AuNPs solution and mixed, then different colors of the mixture appeared.

As shown in Figure 5, 10^2^–10^8^ CFU/mL *Aspergillus flavus* spore suspension and AuNPs solution were mixed, with the concentration increase of *Aspergillus flavus* spore suspension, the mixture color changed from pink to gray gradually. Since the color of AuNPs solution will change after *Aspergillus flavus* is combined with the AuNPs, the greater the color change, the better the combination between the *Aspergillus flavus* and the AuNPs. The result showed that with the increase of the concentration of *Aspergillus flavus* spore suspension, the better the combination of *Aspergillus flavus* spores and AuNPs, the better the enhancement effect of AuNPs solution on the *Aspergillus flavus*.

A significant challenge for many applications of Raman spectroscopy is that the spectra are often accompanied by a strong fluorescence background, especially for biological samples. This background is generally dominated by intrinsic fluorescence from the sample. There is no doubt that the existence of the resonance effect depends on the wavelength of the excitation laser. If the excited photon cannot provide enough energy for the molecule to be in the excited state, the corresponding fluorescence transition will not occur. However, once the fluorescence is generated, its intensity will be much greater than the Raman scattering light, thus masking the characteristics of the Raman signal. Therefore, choosing laser wavelength is an effective way to avoid fluorescent radiation.

The SERS spectra of *Aspergillus flavus* with different concentrations were determined by using a portable Raman spectrometer with a 785 nm laser under the conditions of 300 mW of laser power, 20 s of integration time, and three integration times. According to flocculation theory and hot-spot effect [48,49], when the AuNPs are close to a certain distance, in the gap between particles a highly enhanced electromagnetic field will be formed, and hot spots with excellent enhancement effects will be formed, resulting in strong Raman enhancement signal with the maximum enhancement factor possibly up to 10^14^–10^15^. When the analyte concentration is high, the high concentration ratio AuNPs provides sufficient adsorption sites for the analyte molecules and flocculates to obtain a strong SERS signal. As shown in Figure 6, the average SERS spectra of *Aspergillus flavus* with different concentrations were obtained after baseline correction, normalization and smoothing. The results showed that *Aspergillus flavus* with different concentrations had similar SERS fingerprints at 400–1800 cm^−1^, which were mainly reflected in the characteristic peak absorption and spectral shape, especially at the range of 600–800 cm^−1^, 1200–1400 cm^−1^, and 1500–1600 cm^−1^. Moreover, the intensity of the Raman characteristic peak at 1536 cm^−1^ and in the range of 1200–1400 cm^−1^ increased gradually with the increasing concentration of *Aspergillus flavus* spore suspension. In addition, it also showed that the combination of *Aspergillus flavus* spores and AuNPs was more sufficient with the increase of *Aspergillus flavus* spore concentration and the enhancement effect was better, which was consistent with the results shown in Figure 6. Probably due to the limited hot-spot, the Raman signal does not increase with the concentration of analyte molecules. Even if the concentration ratio of *Aspergillus flavus* is further increased, the AgNPs will not combine with more *Aspergillus flavus*, and the SERS signal will no longer increase. Therefore, 10^2^–10^8^ CFU/mL *Aspergillus flavus* spore suspension should be selected to enable the AgNPs to absorb the abundant *Aspergillus flavus* to generate the Raman signals for quantitative analysis.

The dynamic three-dimensional Raman spectrum revealed a dynamic result of the determination of *Aspergillus flavus* spore suspension added to the AuNPs solution and combined with the AuNPs, as shown in Figure 7. The results showed that the combination state of *Aspergillus flavus* and AuNPs tended to be stable after *Aspergillus flavus* was added to the AuNPs solution for about 10 min. Therefore, the determination effect of the SERS spectrum would be better after 10 min of sufficient combination of *Aspergillus flavus* and AuNPs.

The Raman peak at 1536 cm^−1^ in the SERS diagram of *Aspergillus flavus* was selected as the characteristic peak to carry on the semi-quantitative analysis. According to the signal intensity, the relationship between different concentrations of *Aspergillus flavus* spore suspension and Raman peak intensity at 1536 cm^−1^ was drawn by using the least square fitting method, as shown in Figure 8, indicating that the method could be used for the determination of *Aspergillus flavus.* According to the experimental results, when *Aspergillus flavus* was between 10^2^–10^5^ cfu/mL, the SERS signal intensity did not change significantly with the increase in concentration. Therefore, it could be considered that 10^2^ cfu/mL was the lowest detectable concentration in this detection. Considering the dilution effect of the AuNPs solution on the solution of *Aspergillus flavus*, 20 μL of *Aspergillus flavus* was diluted in 500 μL of AuNPs solution. Therefore, the limit of detection (LOD) was 3.85 cfu/mL.

### 2.3. SERS of Aflatoxin B_1_

The Raman spectra of solid aflatoxin B_1_ were determined so as to reduce the interference of solvents and other factors. It can be seen from Figure 9 that there were many Raman peaks of solid aflatoxin B_1_, and the most obvious characteristic peaks were produced by inelastic scattering between incident laser and aflatoxin B_1_. The aflatoxin B_1_ molecule contains an oxanaphthalene *o*-ketone, a difuran ring, and a cyclopentene ring. The SERS characteristic peaks of aflatoxin B_1_ are 662, 686, 717, 776, 829, 927, 1007, 1082, 1135, 1247, 1309, 1366, 1481, 1554, 1559, 1628, 1691, and 1760 cm^−1^. Among them, the pyran ring respiratory vibration is at 686 cm^−1^, the C–O–C Tensile vibration is at 1247 cm^−1^, and the C–O–C Tensile vibration is at 1309 cm^−1^; 1554 cm^−1^ is the C–C Tensile vibration, and 1599 cm^−1^ is the C–H plane vibration [50,51].

However, due to the large interference of grain surface and interior in the determination of aflatoxin in the process of practical application, as aflatoxin exists in the interior of grain, a certain pretreatment was needed to extract aflatoxin B_1_. Therefore, in the later experiments, aflatoxin B_1_ was dissolved in methanol for SERS determination.

A series of different concentrations of aflatoxin B_1_ methanol solution (0.01 mg/L, 0.005 mg/L, 0.003 mg/L, 0.001 mg/L, and 0.0005 mg/L) were prepared for SERS detection. The non-SERS and SERS of aflatoxin B_1_ were shown in Figure 10, The results showed that the SERS signals of aflatoxin B_1_ would increase with the coupling with AuNPs, while under non-SERS conditions, the Raman signals of normal aflatoxin B_1_ could hardly be detected by portable Raman spectrometer. However, the SERS of aflatoxin B_1_ had fingerprints at 400–1800 cm^−1^, which were mainly reflected in the characteristic peak absorption and spectral shape. In addition, the SERS spectra of aflatoxin B_1_ were similar to that of solid powder by laser confocal micro Raman spectrometer and the characteristic peak increased gradually with the increase of concentration when the standard concentration was in the range of 0.0005–0.01 mg/L. Compared with other peaks, the characteristic peak intensity at 1556 cm^−1^ was more linear with the change of concentrations, so the characteristic peak at 1556 cm^−1^ was selected for the data calculation of detection limit and repeatability. The fitting equation of the curve was shown in Figure 11. Compared with the results of HPLC determination in the published paper carried out by the same research group [52], SERS determination showed the liner range was 0.0005–0.01 mg/L and the limit of detection was 0.0005 mg/L.

## 3. Materials and Methods

### 3.1. Reagents and Materials

Colloidal AuNPs mainly at 60 nm particle size, prepared and provided by our research group according to the primary work published article, were used as the enhanced substrate to magnify the Raman signals [47]. Grain crops (corn) were purchased from the Songyuli farmers’ market of Beijing (China). *Aspergillus flavus* was provided by the Yanjing beer company of Beijing (China). Aflatoxin B_1_ was purchased from Sigma-Aldrich (Shanghai) Trading Co. Ltd. (Shanghai, China).Potato glucose agar medium (PDA) was purchased from Beijing Luqiao Technology Co., Ltd. (Beijing, China). Ultrapure water (18.2 MΩ•cm) was prepared by Millipore (Direct-Q 8 UV-R) and used to prepare all aqueous solutions. The glassware used in the experiment was cleaned with aqua regia (HCl:HNO_3_ = 3:1, *v*/*v*), thoroughly rinsed in water, and dried in an oven at 100 °C prior to use.

### 3.2. Culture of Aspergillus flavus

Next, 8.2 g of PDA medium was placed into a 500 mL conical flask with 200 mL of ultrapure water to obtain the PDA medium solution, which was then heated in a water bath until the medium was clear and transparent. The solution was then placed in a sterilization pot at 121 °C for 20 min. Further, 15 mL of sterilized PDA medium solution was transferred to a disposable dish, then *Aspergillus flavus* was inoculated on the medium and cultured in a constant temperature incubator at 30 °C. After culturing in the incubator for 3–5 days, the mature standard strain was removed and placed in an ultra-clean workbench. The spores on the culture medium were washed with sterile water and filtered to obtain a spore suspension, which was inoculated on corns and cultured at 30 °C.

### 3.3. Preparation of Aflatoxin B_1_ Samples

For the preparation of aflatoxin B_1_ samples, 1.0 mg of Aflatoxin B_1_ powder was accurately weighed with an analytical balance and was transferred to a 1000 mL volumetric flask to obtain a 1.0-mg/L Aflatoxin B_1_ Standard solution after being fixed to the scale with chromatographic methanol. A series of different concentrations of aflatoxin B_1_ methanol solution (0.01 mg/L, 0.005 mg/L, 0.003 mg/L, 0.001 mg/L, and 0.0005 mg/L) was prepared by diluting 1.0 mg/L aflatoxin B_1_ solution for SERS determination.

### 3.4. SERS Measurement and Spectra Preprocessing

The SERS measurement was conducted by using a portable Raman Spectrometer (RamTracer–200, Suzhou OptoTrace Technologies Co. Ltd., Suzhou, China) equipped with a diode laser with a wavelength of 785 nm and power of 300 mW. The wave number range was 250–1800 cm^−1^, the resolution was 2 cm^−1^, and the laser bandwidth was 0.2 nm. The laser power was 300 mW with an integration time of 20 s and was integrated three times.

Before each measurement, X-axis calibration was carried out with acetonitrile (Spectrally Pure) to ensure that the experimental conditions were consistent. Then, 500 μL colloidal AuNPs solution was transferred with a micropipette and placed into a 2.0 mL glass bottle, and 20 μL of *Aspergillus flavus* solution was added and mixed with a vibrator for Raman spectrum measurement immediately. The original spectra were sequentially pre-processed by removing cosmic rays and by baseline correction using Wire 4.1 software (in Via, Renishaw, Gloucestershire, London UK). All of the figures were plotted with Origin software (version 8.0, OriginLab, Northampton, MA, USA).

## 4. Conclusions

In this work, the rapid detection of *Aspergillus flavus* and quantification of Aflatoxin B_1_ in grain crops were presented by using a portable Raman spectrometer combined with AuNPs. The SERS of *Aspergillus flavus* had fingerprints at 400–1800 cm^−1^, which were mainly reflected in the characteristic peak absorption and spectral shape, especially at the range of 600–800 cm^−1^, 1200–1400 cm^−1^, and 1500–1600 cm^−1^. However, the intensity of the Raman characteristic peak at 1536 cm^−1^ increased obviously with the growth of culture time of *Aspergillus flavus* spore suspension. When *Aspergillus flavus* was inoculated on corn grains, the SERS of *Aspergillus flavus* was similar to that inoculated on the medium. With the increase of the concentration of *Aspergillus flavus* spore suspension in the range of 10^2^–10^8^ CFU/mL, the better the combination of *Aspergillus flavus* spores and AuNPs, the better the enhancement effect of AuNPs solution on the *Aspergillus flavus*. A series of different concentrations of aflatoxin B_1_ methanol solution combined with AuNPs were determined based on SERS and their spectra were similar to that of solid powder. Moreover, the characteristic peak increased gradually with the increase of concentration in the range of 0.0005–0.01 mg/L.

The results showed that rapid detection of *Aspergillus flavus* and quantitative determination of aflatoxin B_1_ by using a portable Raman spectrometer combined with colloidal Au nanoparticles based on SERS was reliable and could be used for the assessment of *Aspergillus flavus* and Aflatoxin B_1_ contaminants in grain crops during storage conditions. Compared with the conventional methods, this rapid detection method can greatly shorten the detection time from several hours or even tens of hours to a few minutes, which is very important in the detection of grain crops. Once the grain crops are contaminated by *Aspergillus flavus* and aflatoxin B_1_, we can quickly take effective measures according to the detection results to avoid causing large economic losses.

## Figures and Tables

**Figure 1 molecules-27-05280-f001:**
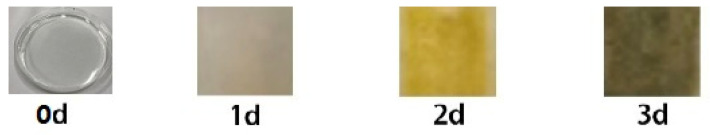
Growth of *Aspergillus flavus* on the medium.

**Figure 2 molecules-27-05280-f002:**
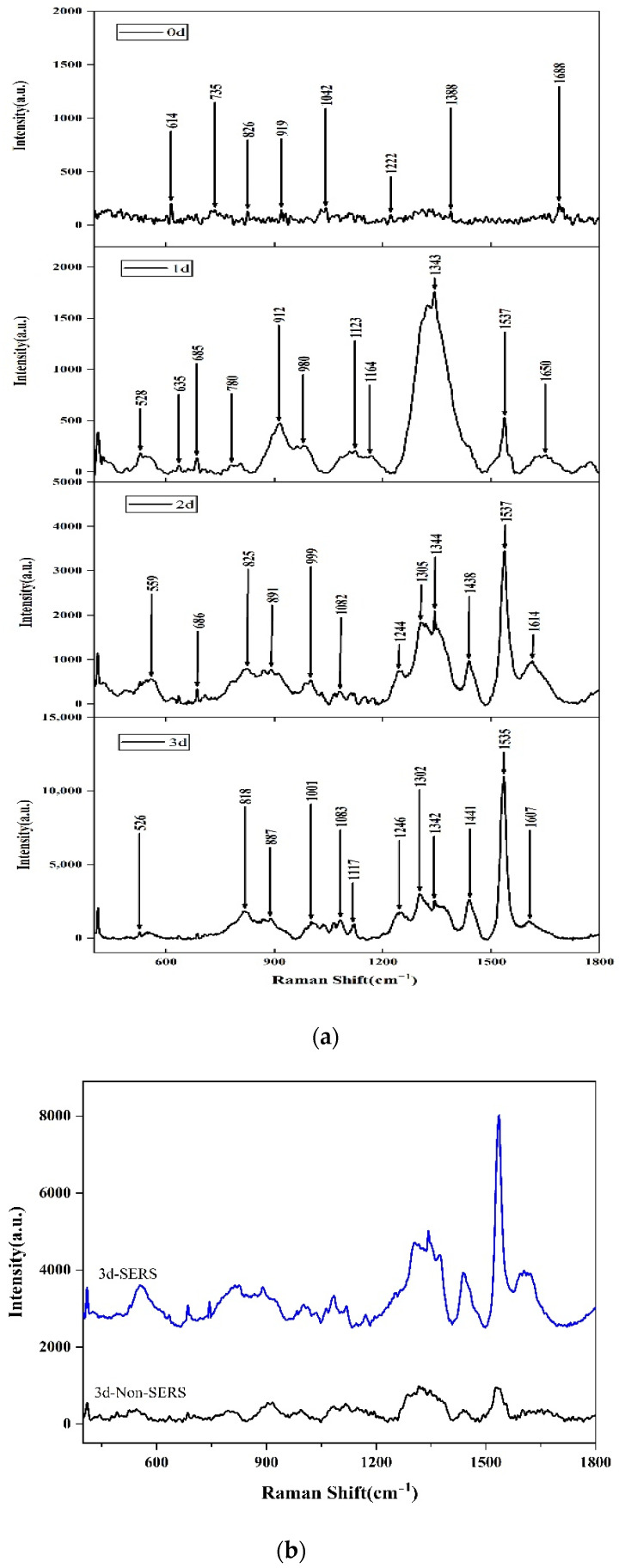
(**a**) SERS of *Aspergillus flavus* on the medium in different culture periods. (**b**) The comparison of non-SERS and SERS of *Aspergillus flavus*.

**Figure 3 molecules-27-05280-f003:**
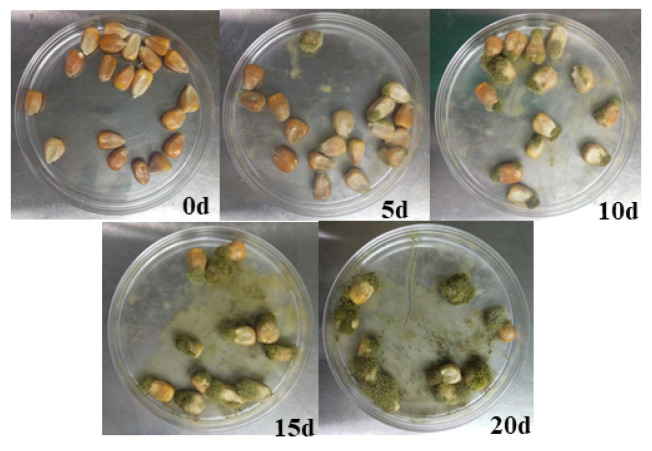
Growth of *Aspergillus flavus* on Corn.

**Figure 4 molecules-27-05280-f004:**
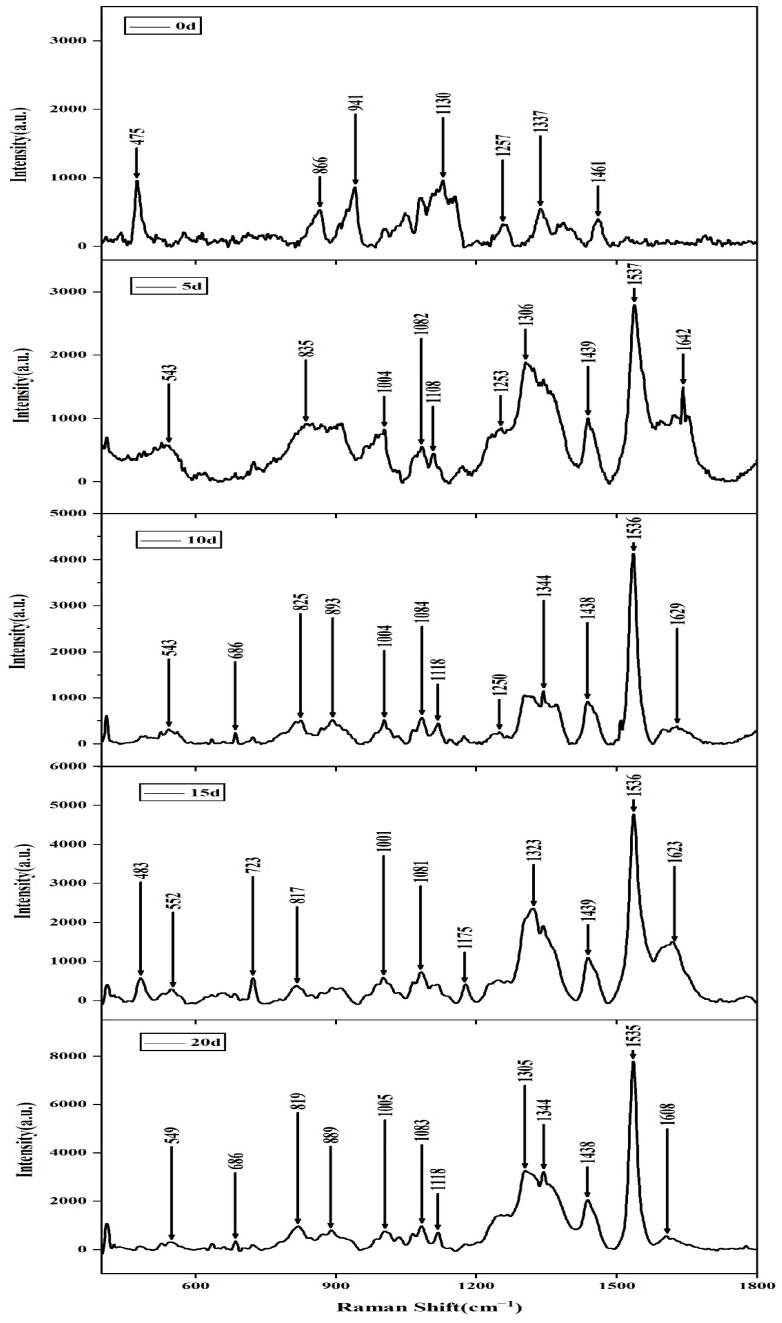
SERS of *Aspergillus flavus* on corn in different culture periods.

**Figure 5 molecules-27-05280-f005:**
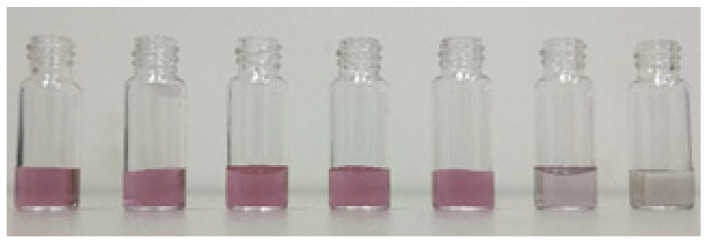
Color change of *Aspergillus flavus* spore suspension mixed with nano gold sol.

**Figure 6 molecules-27-05280-f006:**
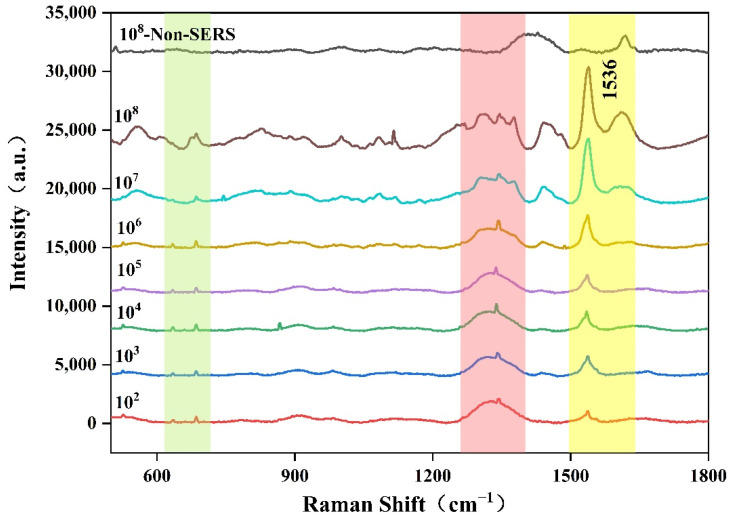
SERS spectra of different concentrations of *Aspergillus flavus* spore suspension.

**Figure 7 molecules-27-05280-f007:**
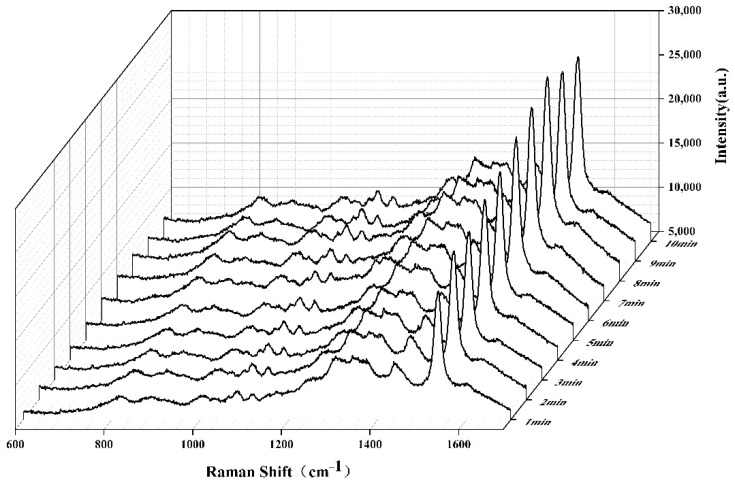
SERS spectrum of dynamic determination of *Aspergillus flavus*.

**Figure 8 molecules-27-05280-f008:**
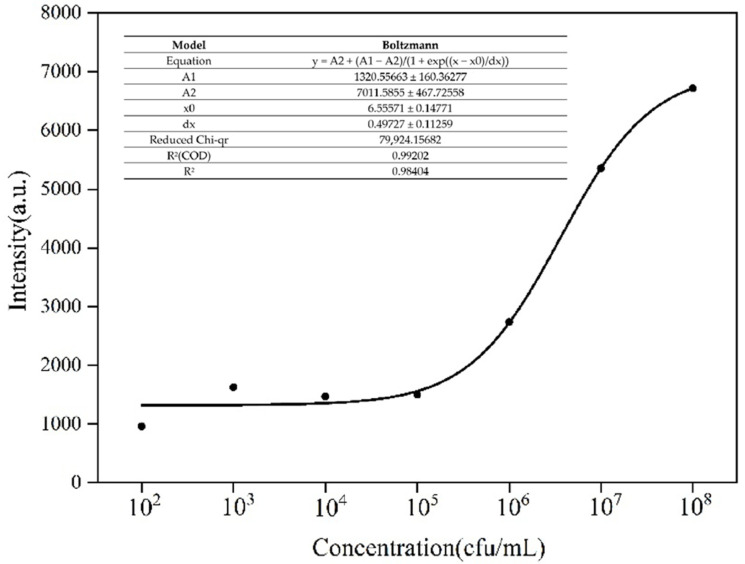
Fitting analysis of SERS test results of different concentrations of *Aspergillus flavus* spore Suspension.

**Figure 9 molecules-27-05280-f009:**
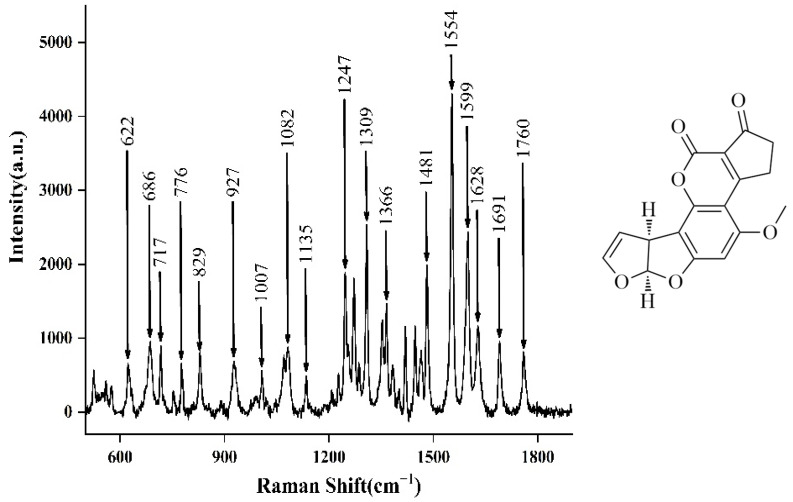
Raman spectrum of aflatoxin B_1_ solid powder determined by laser confocal micro Raman spectrometer.

**Figure 10 molecules-27-05280-f010:**
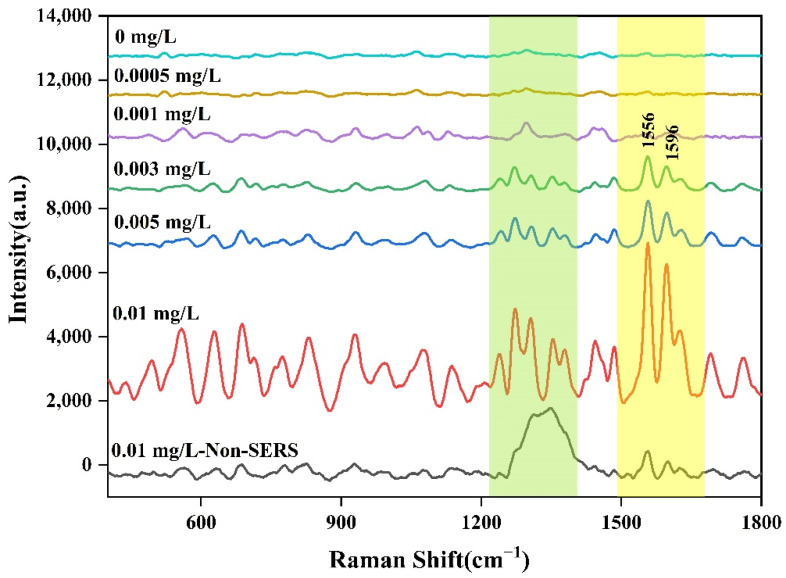
SERS spectra of aflatoxin B_1_ with different concentrations.

**Figure 11 molecules-27-05280-f011:**
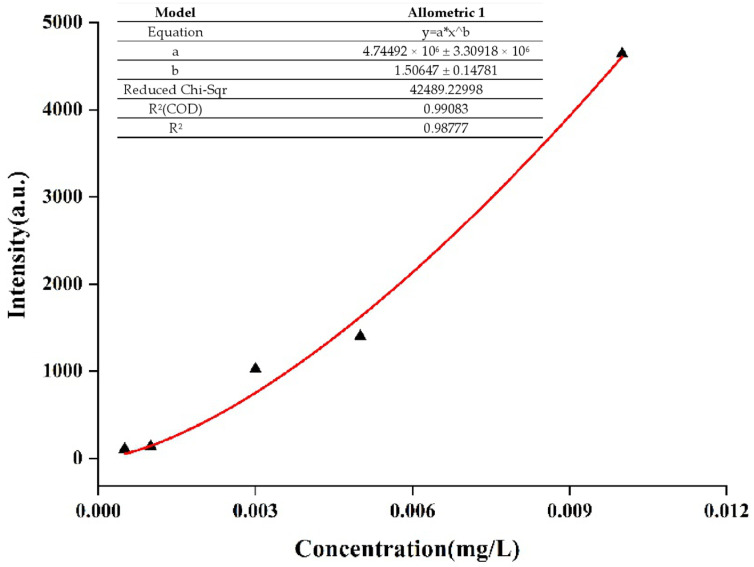
Fitting analysis of SERS test results of different concentrations of aflatoxin B_1_ methanol solution.

## Data Availability

All data are contained within the article.

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
