# Peer review of "Rapid Detection of Aspergillus flavus and Quantitative Determination of Aflatoxin B1 in Grain Crops Using a Portable Raman Spectrometer Combined with Colloidal Au Nanoparticles"

_molecules, 2022, doi:10.3390/molecules27165280_

Round 1

Reviewer 1 Report

Dear All,

The manuscript entitled "Rapid detection of Aspergillus flavus and quantitative determination of aflatoxin B1 in grain crops using a portable Raman spectrometer combined with colloidal Au nanoparticles", by Hui-Qin Wang, Meng-Jia Liu, Yu-Miao Zhang, Hui-Min Zhao, Wen-Jing Lu, Tai-Feng Lin, Ping Zhang and Da-Wei Zheng is an original research paper devoted to a rapid detection of Aspergillus flavus and Aflatoxin B1 by SERS spectroscopy with colloidal Au nanoparticles using a portable Raman spectrometer. Food safety research and the detection of pathogenic fungus are continually interesting topics and may help to avoid causing large economic losses.

The paper describes SERS detection of Aspergillus flavus and Aflatoxin B1 using a portable Raman spectrometer. Analytes under study were irradiated by 785 nm laser source during 20 seconds. There was a study of different culture periods, concentration dependence, time dynamic of SERS spectra (for better understanding of enough time since the Aspergillus flavus was added to the AuNPs solution to achieve the spectra stability), dependences of SERS spectra intensity on analytes concentrations. The SERS characteristic peaks of both Aspergillus flavus and Aflatoxin B1 are clearly seen. The paper is of interest to the research community in the field of biology, sensorics and can be enough appropriate for the Molecules Journal if answered the questions below. For future research, it is useful to use linkers for more selective localization of analyte molecules in the hot spots area.

1. There is a great description of methods and techniques for the detection and quantification of Aspergillus flavus and Aflatoxins in a paper. Also, there are no review of modern SERS metasurfaces. Comparing with the colloidal solutions, there is greater SERS signal stability and better detection limit may be achieved that allow detecting low concentrations up to single molecules level. There are e-beam lithography, electrochemistry, laser ablation methods. Please work with the introduction. You may review the following papers https://doi.org/10.1038/srep18567, https://doi.org/10.3390/app11041375, https://onlinelibrary.wiley.com/doi/10.1002/adma.201506251, https://doi.org/10.3390/nano9050677, DOI: 10.3390/nano11061394

2. For correct spectra analysis there is a need in both Raman and SERS spectra. Please provide a Raman spectra of Aspergillus flavus. Also, for Aspergillus flavus it is written that there are characteristics picks at 400-1800 cm-1, but there is no peak analysis and explanation.

3. There are no words about fluorescence background (if it was). Please explain how the fluorescence was affected the original SERS spectra.

4. There are calibration curves in the paper (fig. 8 and 11). Based on such curves (the intensity dependence on the analyte concentration) it is possible to calculate the limit of detection (LOD). There is a suggestion in the line 147 that the maximum enhancement factor may be of 1014-15 power, so it will be interesting to find out an experimental LOD.

5. Is a laser power of 30 mW or 300 mW??? (Lines 247, 249)

I think that the article in principle can be accepted for publication in the Journal of Luminescence with major revision, and taking into account the above comments.

Sincerely yours.

Author Response

Point 1: There is a great description of methods and techniques for the detection and quantification of Aspergillus flavus and Aflatoxins in a paper. Also, there are no review of modern SERS metasurfaces. Comparing with the colloidal solutions, there is greater SERS signal stability and better detection limit may be achieved that allow detecting low concentrations up to single molecules level. There are e-beam lithography, electrochemistry, laser ablation methods. Please work with the introduction. You may review the following papers https://doi.org/10.1038/srep18567,  https://doi.org/10.3390/

App11041375, https://onlinelibrary.wiley.com/doi/10.1002/adma.201506251, https://doi.org/10.3390/nano9050677, DOI: 10.3390/nano11061394

Response 1: Thank you for your advice! We added some contents of SERS substrates and seven relevant references (reference 32-38) in the introduction section. Caldwell et al utilized spherical gold nanoparticles with 14 nm and 46 nm diameters to improve the scattering signal obtained during Raman spectroscopy measurements to de-tect small plastic particles. Bharathi et al utilized picosecond laser-ablated gold na-noparticles (Au NPs) as surface-enhanced Raman scattering (SERS) substrates to detect the dye methylene blue and a chemical warfare agent simulant (methyl salicylate). Zavyalova et al provided a SERSaptasensor based on colloidal solutions which combines rapidity and specificity in quantitative determination of SARS-CoV-2 virus. Compar-ing with the colloidal solutions, there is greater SERS signal stability and better detection limit may be achieved that allow detecting low concentrations up to single molecules level based on some SERS substrates which were prepared by the template-assisted electro-deposition, binary-template-assisted electrodeposition, pulsed laser ablation, and other methods . The surface nanostructures of artificially roughened metal thin films display many hot spots making them excellent SERS substrates.

Point 2: For correct spectra analysis there is a need in both Raman and SERS spectra. Please provide a Raman spectra of Aspergillus flavus. Also, for Aspergillus flavus it is written that there are characteristics picks at 400-1800 cm-1, but there is no peak analysis and explanation.

Response 2: Thank you for your advice! The Raman spectra of Aspergillus flavus was supplemented and compared with the surface enhanced Raman spectrum, as shown in the figure 2. The peak of the surface enhanced Raman spectra of Aspergillus flavus was analyzed and explained . The peak at 1605-1615cm-1 represents C=O stretching in proteins, 1343-1346 cm-1 represents DNA base, 1315-1317cm-1 represents vibration of (-C=C-) conjugated of Amine III, 1302-1306 cm-1 represents carbohydrates, and 805-825 cm-1 represents protein respectively.

Point 3: There are no words about fluorescence background (if it was). Please explain how the fluorescence was affected the original SERS spectra.

Response 3: Thank you for your advice! The interference of fluorescent background was supplemented in Section 2.2. A significant challenge for many applications of Raman spectroscopy is that the spectra are often accompanied by a strong fluorescence background, especially for biological samples. This background is generally dominated by intrinsic fluorescence from the sample. There is no doubt that the existence of the resonance effect depends on the wavelength of the excitation laser. If the excited photon cannot provide enough energy for the molecule to be in the excited state, the corresponding fluorescence transition will not occur. However, once the fluorescence is generated, its intensity will be much greater than the Raman scattering light, thus masking the characteristics of the Raman signal. Therefore, choosing laser wavelength is an effective way to avoid fluorescent radiation.

Point 4: There are calibration curves in the paper (fig. 8 and 11). Based on such curves (the intensity dependence on the analyte concentration) it is possible to calculate the limit of detection (LOD). There is a suggestion in the line 147 that the maximum enhancement factor may be of 1014-15 power, so it will be interesting to find out an experimental LOD.

Response 4: Thank you for your suggestion! According to the experimental results, when Aspergillus flavus was between 102-105 cfu/mL, the SERS signal intensity did not change significantly with the increase of concentration. Therefore, it could be considered that 102 cfu/mL was the lowest detectable concentration in this detection. Considering the dilution effect of the AuNPs solution on the solution of Aspergillus flavus, 20uL of Aspergillus flavus was diluted in 500uL of AuNPs solution. Therefore, the limit of detection (LOD) was 3.85 cfu/mL.

In the line 147, the sentence “resulting in strong Raman enhancement signal with 1014~1015 of enhancement factor” was revised to “resulting in strong Raman enhancement signal with the maximum enhancement factor possibly up to1014~1015”.

Point 5: Is a laser power of 30 mW or 300 mW??? (Lines 247, 249)

Response 5: Thank you for your review! It is a writing error. The laser power was 300 mW, and it has been corrected in the paper.

Reviewer 2 Report

This manuscript by Wang et al. reports on detections of Aspergillus flavus and aflatoxin using a portable Raman spectrometer.  The authors obtained Raman spectra of cultured Aspergillus flavus with fingerprinting vibrational peaks at around 1535 cm^-1.  They also added Au nanoparticles to enable SERS detections of the strains, which allowed the detections at a concentration as low as 10^2 CFU/mL.  The SERS approach was also implemented for detecting aflatoxin B1 extracted from the fungi, which demonstrated a detection limit of 0.5 μg/L.

The work is well-organized to prove the ability of the SERS measurements for detecting Aspergillus flavus and aflatoxin.  However, there is not enough information provided to reproduce the experiments.  Some control experiments are also required to support their conclusions.  I will therefore recommend the publication after the authors address the following points as a major revision.

1.       It is not explained how the Raman spectra in Figure 2 were obtained.  It says “SERS of Aspergillus flavus” in the caption but no information about the conditions such as presence or absence of metal particles and the fungi concentrations was provided.

2.       The advantage of adding Au nanoparticles for the detections of Aspergillus flavus and aflatoxin should be shown by data.  For example, the authors can show the Raman spectra of Aspergillus flavus and aflatoxin without adding the Au nanoparticles.

3.       Control experiments are missing to validate that the Raman spectra are surely of Aspergillus flavus or aflatoxin.  As the authors state that there are many contaminants included in their samples, it is necessary to perform the optical measurements on crop samples without Aspergillus flavus.

4.       On page 8, please elaborate what kind of pretreatment was performed to purify aflatoxin.  It is also better mentioned what filters were used for the Aspergillus flavus samples in section 3.

5.       The sensing performance should be compared to the other procedures such as plate counting and electrochemical impedance spectroscopy. 

6.       Please clarify what is shown by the images in Figure 1 and how they were obtained.   

Author Response

Point 1:  It is not explained how the Raman spectra in Figure 2 were obtained.  It says “SERS of Aspergillus flavus” in the caption but no information about the conditions such as presence or absence of metal particles and the fungi concentrations was provided.

Response 1: Thank you for your review! In the 2.1 section, the following contents were supplemened: After centrifuged the culture medium (1 mL) at 8000 g for 5 min, the Aspergillus flavus spores were collected and resuspended in 1 mL of 0.85% sterile normal saline, and then centrifuged and washed under the same conditions. The above procedure was repeated for another 2∼3 times to remove the culture medium and get the sample, of which 20 uL was taken and added to 500 uL of AuNPs solution and mixed for SERS detection. In this section, SERS analysis was only conducted for the changes of Aspergillus flavus on the culture medium that can be seen by the naked eyes. The concentration of Aspergillus flavus was discussed in part 2.2, as shown in Figure 6.

Point 2: The advantage of adding Au nanoparticles for the detections of Aspergillus flavus and aflatoxin should be shown by data.  For example, the authors can show the Raman spectra of Aspergillus flavus and aflatoxin without adding the Au nanoparticles.

Response 2: Thank you for your advice! In this experiment, Aspergillus flavus and aflatoxin B1 without adding AuNPs solution were detected by portable Raman spectroscopy (expressed by non-SERS) respectively. The results showed that the SERS signals of Aspergillus flavus and aflatoxin B1 would increase with the coupling with AuNPs, while under non-SERS conditions, the Raman signals of normal Aspergillus flavus and aflatoxin B1 could hardly be detected by portable Raman spectrometer. non-SERS spectra of Aspergillus flavus and aflatoxin B1 were supplemened, as shown in figure 2 and figure 10.

Point 3: Control experiments are missing to validate that the Raman spectra are surely of Aspergillus flavus or aflatoxin.  As the authors state that there are many contaminants included in their samples, it is necessary to perform the optical measurements on crop samples without Aspergillus flavus.

Response 3: Thank you for your advice! In this study, some experiments about SERS measurements on crop samples without Aspergillus flavus and aflatoxin B1 were performed. Fresh culture medium and corns, without Aspergillus flavus and aflatoxin B1, were called blank samples for SERS determination after the same treatment as the positive samples under the same conditions. The results showed that the SERS signals of Aspergillus flavus and aflatoxin B1 in black samples could hardly be detected by portable Raman spectrometer. On the contrary, the SERS signals of positive samples were obviously easy to be obtained. The SERS spectra of Aspergillus flavus and aflatoxin B1 in black samples were supplemened, as shown in figure 1, figure 2, figure 4 and figure 10.

Point 4: On page 8, please elaborate what kind of pretreatment was performed to purify aflatoxin.  It is also better mentioned what filters were used for the Aspergillus flavus samples in section 3.

Response 4: Thank you for your suggestion! In the section 3, preparation of Aflatoxin B1 samples was supplemened. In this paper, only a series of different concentrations of aflatoxin B1 methanol solution were determined to discuss the method and quantification. The grain crops were not involved, so the preparation method of the standard solution was introduced. The filtration treatment in section 3 was used to obtain the spore suspension of Aspergillus flavus and remove the culture medium during the sample processing.

Point 5: The sensing performance should be compared to the other procedures such as plate counting and electrochemical impedance spectroscopy.

Response 5: Thank you for your advice! After a series of different concentrations of aflatoxin B1 methanol solution were determined by portable Raman spectrometer, SERS determination showed the liner range was in 0.0005-0.01 mg/L and the limit of detection was 0.0005 mg/L. Compared with the linear range of 0.1-0.5 mg/L by HPLC determination in the published paper carried out by the same research group, the linear range of 0.0005-0.01 mg/L was improved a wider degree.

Point 6:  Please clarify what is shown by the images in Figure 1 and how they were obtained.

Response 6: Thank you for your review! When Aspergillus flavus was inoculated on the culture medium, the increase of Aspergillus flavus on the culture medium could be clearly observed with the naked eyes with the increase of culture days. Figure 1 was obtained from a conventional camera, and the purpose was to make a preliminary detection with a portable Raman spectrometer when changes of Aspergillus flavus were visible to the naked eyes.

Round 2

Reviewer 2 Report

The authors have replied to all of my comments.   I now recommend publication of this manuscript.